# The PERK Branch of the Unfolded Protein Response Promotes DLL4 Expression by Activating an Alternative Translation Mechanism

**DOI:** 10.3390/cancers11020142

**Published:** 2019-01-25

**Authors:** Manon Jaud, Céline Philippe, Loic Van Den Berghe, Christèle Ségura, Laurent Mazzolini, Stéphane Pyronnet, Henrik Laurell, Christian Touriol

**Affiliations:** 1Inserm UMR1037, CRCT (Cancer Research Center of Toulouse), CNRS ERL5294, Université Toulouse III Paul-Sabatier, F-31037 Toulouse, France; manon.jaud@inserm.fr (M.J.); c.philippe@qmul.ac.uk (C.P.); Loic.Vandenberghe@inserm.fr (L.V.D.B.); christele.segura@inserm.fr (C.S.); laurent.mazzolini@inserm.fr (L.M.); stephane.pyronnet@inserm.fr (S.P.); 2Vectorology Plateform, Technological pole CRCT, F-31037 Toulouse, France; 3Inserm UMR1048, I2MC (Institut des Maladies Métaboliques et Cardiovasculaires), Toulouse, France; henrik.laurell@inserm.fr

**Keywords:** DLL4 (delta like ligand 4), angiogenesis, IRES (internal ribosome entry site), hypoxia, endoplasmic reticulum stress, UPR (unfolded protein response), PERK (PKR-Like endoplasmic reticulum kinase)

## Abstract

Delta-like 4 (DLL4) is a pivotal endothelium specific Notch ligand that has been shown to function as a regulating factor during physiological and pathological angiogenesis. DLL4 functions as a negative regulator of angiogenic branching and sprouting. Interestingly, *Dll4* is with *Vegf-a* one of the few examples of haplo-insufficiency, resulting in obvious vascular abnormalities and in embryonic lethality. These striking phenotypes are a proof of concept of the crucial role played by the bioavailability of VEGF and DLL4 during vessel patterning and that there must be a very fine-tuning of DLL4 expression level. However, to date the expression regulation of this factor was poorly studied. In this study, we showed that the *DLL4* 5′-UTR harbors an Internal Ribosomal Entry Site (IRES) that, in contrast to cap-dependent translation, was efficiently utilized in cells subjected to several stresses including hypoxia and endoplasmic reticulum stress (ER stress). We identified PERK, a kinase activated by ER stress, as the driver of *DLL4* IRES-mediated translation, and hnRNP-A1 as an IRES-Trans-Acting Factor (ITAF) participating in the IRES-dependent translation of DLL4 during endoplasmic reticulum stress. The presence of a stress responsive internal ribosome entry site in the DLL4 msRNA suggests that the process of alternative translation initiation, by controlling the expression of this factor, could have a crucial role in the control of endothelial tip cell function.

## 1. Introduction

Coordinately expressed and regulated genes control most physiological processes. This is the case for angiogenesis, the process of the expansion of existing blood vessel mainly by sprouting new branches from pre-existing blood vessels and leading to the outgrowth of new capillaries to form a new functional vascular network. 

Angiogenesis is critical for many physiological processes such as embryonic development, wound healing, or vessel penetration into avascular regions and in pathological states including retinopathy, chronic inflammatory disorders (e.g., psoriasis or rheumatoid arthritis) and of course solid tumor development. Angiogenesis depends on the highly coordinated actions of a variety of pro-angiogenic regulators, the most prominent and best characterized being Vascular Endothelial Growth Factor A (VEGF-A) and Fibroblast Growth Factor 2 (FGF-2) which were among the first pro-angiogenic molecules to be identified [1].

Among the stimuli responsible for the up-regulation of pro-angiogenic factors, hypoxia has been of particular interest because of its role in cancer progression. It is clearly established that VEGF-A and FGF-2 are strongly expressed in hypoxic tissues, allowing the recruitment of new blood vessels from surrounding territories to ensure their needs of oxygen and nutrients. Hypoxia contributes to angiogenesis by transcriptionally activating several angiogenic factors as well as their receptors thus facilitating the recruitment of endothelial cells to the site of hypoxia, but VEGF-A and FGF-2 expression is also up-regulated under hypoxic conditions [2,3] and after ER stress [4] through Internal Ribosome Entry Site (IRES)-mediated translation [5]. After tissue reoxygenation, the expression of angiogenic factors decreases. This elegant negative feedback mechanism is a key event in the regulation of blood vessel growth.

However, the vascular response to angiogenic factors is also dependent on other regulatory mechanisms. Several studies demonstrate that one such regulator essential for tumor neovascularization is the NOTCH ligand DLL4, which is one of three delta-like ligands in the mammalian genome [6]. DLL4 is an important component of the NOTCH pathway, which is critical for embryonic vascular development and arterial specification. It is weakly expressed in adult tissues, but markedly induced in murine and human tumor vasculature [7,8,9]. Using different experimental models and a variety of genetic and pharmacologic approaches, several studies report that the DLL4/NOTCH pathway is a critical negative regulator of tumor angiogenesis, acting to restrain excessive VEGF-induced vascular sprouting and angiogenesis. The DLL4 pathway regulates sprouting and branching behaviors by influencing the formation of vascular ‘tip cells’—specialized endothelial cells at the leading edge of vascular sprouts [10,11,12,13,14,15,16]. The tip cells which express high levels of VEGFR-2, VEGFR-3 and platelet-derived growth factor-B, are characterized by their cellular protrusions or filopodia that sense the local environment and migratory behavior to guide the growth of new blood vessels towards the source of angiogenic growth factors [17]. VEGFR2 signaling induces the expression of the Notch ligand DLL4 on the surface of the tip cell membrane allowing the suppression of tip-cell features in adjacent stalk cells via DLL4/NOTCH-mediated lateral inhibition [18]. Then the endothelial stalk cells follow the polarized migration of tip cells and proliferate in order to form new blood vessels.

During this process, endothelial cells are exposed to gradient of oxygen [19]. Indeed, since lumen formation of new vessel has not occurred, blood flow is not present, thus endothelial cells from these new blood vessels are under hypoxia, especially the tip cells due to their position furthest away from the circulating blood. It was demonstrated that *Dll4* haploinsufficiency causes embryonic lethality and reduces tumor growth due to defects in the development of the vasculature [7,20]. These striking phenotypes resulting from heterozygous deletion of *Dll4* indicate that vascular development may be very sensitive to subtle changes in *Dll4* expression. Interestingly VEGF-A and DLL4 are the only proteins for which the heterozygosity results in a lethal embryonic phenotype and obvious vascular abnormalities, highlighting the essential and unique role of both proteins during angiogenesis [7,21,22].

Furthermore, several data indicate that VEGF-A and DLL4 are coordinately expressed and that VEGF activates DLL4 expression via VEGFR2 signaling [15,18,23,24]. Moreover, in vitro, hypoxia can induce transcription activation of both *Vegf-a* and *Dll4* in endothelial cells [25]. Finally, endothelial expression of DLL4 was demonstrated to be significantly associated with VEGF-A in many cancers including glioma, colon, nasopharyngeal and lung cancers [26,27,28,29]. Taken together, these data indicate a potential co-regulation of these two genes.

It is well known that DLL4 expression is regulated by transcriptional and post-transcriptional (i.e., 3′-end processing) mechanisms, but the translational regulation of DLL4 messenger has not yet been studied. In this study, we sought to further investigate the role of stress responses in DLL4 expression regulation. We have identified an Internal Ribosome Entry Site (IRES) in the 5′-UTR of DLL4 mRNA which is activated under hypoxic and ER stress conditions. Further, we have identified PKR-like ER-associated protein kinase (PERK), a kinase activated during ER stress which phosphorylates the eIF2α subunit and impairs the generation of the ternary complex Met-tRNAi-eIF2-GTP, as the main regulator of DLL4 IRES-mediated translation and hnRNPA1 as an IRES Trans Acting Factor regulating DLL4 IRES-mediated translation during stress.

## 2. Results

### 2.1. The Human DLL4 Transcript Contains a Functional IRES Which is Active in Different Cell Types

Strong conservation of non-coding exonic sequences during vertebrates’ evolution often means involvement in post-transcriptional regulation of gene expression [30]. Interestingly, *DLL*4 5′-UTR shows fairly high conservation, with more than 70% mRNA sequence identities between 14 mammal species (Appendix AA). This indicates that this non-coding region may contain functional RNA structures or regulatory sequences important for the translation of the *DLL4* mRNA.

Indeed, according to the prediction, the 320-nucleotide-long human *DLL4* 5′-UTR is highly structured (Appendix AB) due to a high percentage of G and C residues (more than 70%). Analysis by the MFold prediction algorithm revealed that the full human *DLL4* 5′-UTR form a free energy structure of less than -150 kcal/mol, but also that the first 85 bases of the human DLL4 mRNA might form a very stable secondary structure (ΔG = −37.2 kcal/mol) (Appendix AB). Secondary structural features of the mRNA 5′-untranslated region (UTR) are important for translational regulation by affecting the recruitment and positioning of the ribosome at a favorable initiation codon [31,32], and both thermal stability and cap-to-hairpin proximity affect translational efficiency, particularly when the predicted hairpin stability is below −25 kcal/mol [33]. These structural elements act as strong barriers to scanning ribosomes in the 5′-UTR of mRNAs and are incompatible with the conventional scanning initiation model. In general, the corresponding mRNAs have evolved nonconventional mechanisms to initiate translation, including IRES elements, which are specialized RNA regulatory sequences governing cap-independent translational initiation. A typical example is the Ornithine Decarboxylase (ODC) mRNA, which possesses, in the cap proximal part, a very stable stem-loop structure highly inhibitory of cap-dependent translation [34] but also an IRES element allowing an efficient translation of this mRNA [35].

Thus, we first investigated whether the *DLL4* 5′-UTR contained any IRES activity. In this aim, a classical bicistronic reporter plasmid was constructed by the insertion of a cDNA corresponding to the *DLL4* 5′-UTR (nts 1 to 320) between two reporter gene sequences, the first encoding Renilla Luciferase which is strictly dependent upon cap-dependent translation, and the second encoding Firefly Luciferase which is dependent upon the presence of an IRES for its translation (Figure 1A) [36,37]. Twenty-four hours after transfection into HUVEC, HeLa and NIH-3T3 cells, Renilla and Firefly activities were measured and the LucF (Firefly)/LucR (Renilla) ratios were calculated as an index of IRES/Cap-dependent translation (Figure 1B–D). The EMCV and VEGF constructs containing IRES were used as positive controls [4,38], and the FGF-1A construct was used as a negative control since although it contains an IRES, its activity is cell type-dependent [39]. Our results showed that *DLL4* 5′-UTR contained a putative IRES with an activity comparable to that of *VEGF-A* in the three cell lines tested (Figure 1B–D). To rule out the possibility of the presence of a cryptic promoter in the *DLL4* 5′-UTR, which could also cause Firefly activity, we cloned the *DLL4* 5′-UTR in the Tet-Off bicistronic system [2]. In this system, the bicistronic cassette is under the control of a doxycycline-repressible promoter (Figure 1E). Thus the LucF/LucR ratio is expected to be stable and independent of the CMV-driven expression level if expression of the second cistron is IRES dependent. In contrast, if there is an intercistronic cryptic promoter, LucF expression will be independent of the CMV promoter, and thus the LucF/LucR ratio will increase proportionally to the repression of CMV promoter by doxycycline. The individual two *VEGF* IRESes, known to be promoterless, and the full length 5′-*VEGF-A* 5′-UTR, known to contain a promoter, were used as controls [2]. The results clearly show that the LucF/LucR ratio increased with the inactivation of the CMV/Tet promoter after doxycycline treatment when the full *VEGF-A* 5′-UTR was present between the two cistrons. In contrast, the ratios remained unchanged when the individual *VEGF* IRESes or the *DLL4* 5′-UTR were tested, demonstrating the absence of cryptic promoter in the intercistronic region. Altogether these results indicate that the *DLL4* 5′-UTR possesses an IRES that can initiate cap-independent translation.

### 2.2. DLL4 IRES Activity is Stimulated by Hypoxia

Given that DLL4 is mainly expressed by the tip cells, localized at the leading edge of vessel sprouts in an unfavorable hypoxic microenvironment, we next tested the effect of hypoxia on the *DLL4* IRES activity (Figure 2). Bicistronic constructs were transfected in HeLa and HUVEC cells. As expected, under hypoxic conditions HIF1α expression increased and Eukaryotic Initiation Factor 4E-Binding Protein 1 (4E-BP1) was dephosphorylated (shift to a band of lower apparent molecular weight in western blot) (Figure 2A,C). In the meantime we observed the inhibition of cap-dependent translation (decreased LucR activity) and no effect on LucF activity driven by *EMCV* IRES, while the *DLL4* IRES-driven expression of LucF increased in both cell types after hypoxia (Figure 2B,D). Hypoxic stress resulted in a 2.5-fold stimulation of *DLL4* IRES activity whereas *EMCV* IRES-mediated translation (LucF activity) remains stable in the two cell types (Figure 2B,D).

These results confirm the presence of a bona fide IRES element in the 5′-UTR of DLL4 mRNA, which is activated under hypoxia.

### 2.3. The DLL4 IRES is Stimulated Following ER Stress

Many cellular stresses, including hypoxia, can activate ER dependent pathways by inducing an accumulation of misfolded/unfolded proteins within the ER [4,40]. In order to investigate whether the *DLL4* IRES responds to ER stress, we transfected HeLa cells with constructs containing the *EMCV*, or the *DLL4* IRES and treated them for 4 h with increasing amounts of dithiothreitol (DTT) a well-known ER stress inducer (Figure 3). To confirm ER stress activation we verified the increased levels of both *XBP1* splicing by RT-PCR and phospho-eIF2α (p-eIF2α) by western blotting after DTT treatment (Figure 3A). By comparing the ratio of luciferase reporter activities (LucF/LucR) between treated and control cells, we found that the relative *DLL4* IRES activity was around 5-fold greater in cells treated with 8 mM DTT for 4 h versus control cells (Figure 3B). On the contrary, and as previously described, the *EMCV* IRES was not activated under the same conditions [4,40]. As expected, triggering of ER stress resulted in a decrease of LucR expression in a dose-dependent manner after DTT, given its inhibitory effect on cap-dependent translation [4,40]. Interestingly, whereas the *EMCV* IRES-mediated expression of the second cistron encoding LucF also diminished dose-dependently, the LucF expression driven by the *DLL4* IRES increased after DTT treatment (Figure 3B). These results demonstrate that the *DLL4* IRES is activated by ER stress. Furthermore, during ER stress, DLL4 cap-independent translation is increased when global cap-dependent translation is repressed.

### 2.4. PERK Kinase is Required for DLL4 IRES-Mediated Translational Upregulation During ER Stress in Vitro

In response to ER stress, cells activate the physiological unfolded protein response (UPR) triggered by the activation of three ER transmembrane sensors: PKR-like ER-associated protein kinase (PERK), Activating Transcription Factor-6 (ATF6) and Inositol-Requiring Enzyme-1 (IRE1) [41,42]. To investigate the pathways involved in *DLL4* IRES activation during ER stress, we down-regulated the expression of these three ER stress sensors in HeLa cells. Transient downregulation of PERK, ATF6 and IRE1 by siRNA interference was confirmed by western blotting or semi-quantitative RT-PCR (Figure 4A). After DTT treatment, the phosphorylation of eIF2α was, as expected, diminished in cells transfected with PERK siRNA as was XBP-1 splicing after transfection of IRE1 siRNA (Figure 4B). We then co-transfected the bicistronic constructs with the respective siRNAs and calculated the ratio of IRES activities (LucF/LucR) between cells treated or not with DTT. The down-regulation of the three ER-stress transducers had no effect on *EMCV* IRES activity after DTT treatment (Figure 4C). On the other hand, ER stress-induced stimulation of the *DLL4* IRES was only affected after PERK down-regulation, suggesting that PERK, but not IRE1 or ATF6, is required for the control of *DLL4* IRES activity.

Similar results were obtained after both specific pharmacological PERK activation with CCT020312 or inhibition with GSK2606414 during ER stress (Figure 4D–F). As expected, PERK inhibition by GSK2606414, which had no effect on the *XBP1* splicing efficiency, prevented DTT-induced eIF2α phosphorylation (Figure 4D), whilst PERK activation by CCT020312, which does not initiate cytoplasmic *XBP1* splicing as opposed to DTT, stimulated eIF2α phosphorylation (Figure 4E). Consistently, the inhibition or activation of PERK affected the *DLL4* IRES (Figure 4F), whereas no effect on *EMCV* IRES activity was observed. Taken together, these results indicate that PERK activation is sufficient to stimulate *DLL4* IRES activity.

To complement the pharmacological approach, we used an already described inducible tetracycline/leucine zipper-based dimerization system enabling artificial activation of the PERK pathway [4,40]. In this model the intraluminal ER sensor domain of PERK is replaced by a c-Jun leucine zipper fused to a HA tag (PERK-LZ; Figure 5A). Thus, the addition of increasing concentrations of doxycycline induces specific expression and subsequent dimerization and activation of PERK-LZ as visualized by concomitant expression of the PERK-LZ (HA) expression and phosphorylation of eIF2α (p-eIF2α) (Figure 5B). These cells were then used to confirm that selective PERK pathway activation was sufficient to stimulate *DLL4* IRES activity. Forty-eight hours after the addition of 1 µg/mL doxycycline to the culture medium, cells were transfected with the bicistronic vectors containing the *EMCV*, and *DLL4* 5′-UTR IRES, and luciferase activities were measured 24 h later. An increase in IRES activity was seen with the *DLL4* 5′-UTR constructs but not with the construct containing the *EMCV* IRES (Figure 5C), further demonstrating that the PERK pathway is sufficient for *DLL4* IRES activation during ER stress.

Finally, to investigate whether PERK alone is sufficient to induce *DLL4* IRES activation or instead if signaling downstream PERK is required, we evaluated the effect of ATF4 (Activating Transcription Factor 4) down-regulation by siRNA on ER stress-induced *DLL4* IRES activation. ATF4 regulates the transcription of a number of genes involved in stress response and cell survival and, in contrast to most transcripts, the translation of ATF4 is enhanced as a consequence of increased phosphorylation of eIF2α. siRNA-mediated ATF4 knockdown impaired neither eIF2α phosphorylation nor PERK activation (shown by supershift in immunoblots of total PERK) (Figure 5D) and had no effect on the stimulation of *DLL4* IRES after induction of ER stress by DTT (Figure 5E). This indicates that activation of the PERK pathway, independently of ATF4, is sufficient to stimulate *DLL4* IRES activity.

To investigate the potential role of eIF2α phosphorylation in this process we transfected bicistronic vectors in mouse embryo fibroblasts (MEFs) derived from either wild-type mice or from eIF2α knock-in mice that have a homozygous mutation precluding eIF2α phosphorylation (S51A). As expected, no phosphorylation of eIF2α was observed in mutant MEFs after ER stress induction by DTT (Figure 5F) while this treatment efficiently induced a comparable *XBP1* splicing in both S51A and WT MEFs (Figure 5F). The ratio of Firefly to Renilla luciferase between ER stress inducers treated and control cells remained stable with bicistronic vector containing the *EMCV* IRES and was significantly increased with the *DLL4* IRES only in WT MEFs but not in S51A MEFs (Figure 5G). Taken together these results independently confirm that translation from *DLL4* IRES is stimulated by PERK during ER stress, and demonstrate that phosphorylation of eIF2α is required.

### 2.5. hnRNP A1 Modulates DLL4 IRES-Mediated Translation

IRES-dependent translation efficiency is controlled by RNA-binding proteins known as IRES trans-acting factors (ITAF). Subcellular relocalization of ITAFs plays a crucial role in the modulation of IRES-dependent translation efficiency [43]. Indeed, many RNA-binding proteins are able to shuttle between the nucleus and the cytoplasm. For example, it has been reported that cytoplasmic relocalization of ITAFs, such as hnRNPA1, may either activate or inhibit IRES activity when accumulating in the cytoplasm [44,45]. Interestingly, many stresses like UVC, osmotic shock or ER stress induce cytoplasmic hnRNPA1 relocalization [44,45,46]. Moreover, it was demonstrated that this hnRNPA1 cytoplasmic accumulation, during osmotic stress, requires eIF2α phosphorylation [47].

Thus we evaluated whether hnRNP A1 could play a role on the activation of DLL4 cap-independent translation after ER-stress-mediated eIF2α phosphorylation. After verification of extraction efficiency of cytosolic and nuclear proteins in each fraction (Appendix A), we analyzed the nuclear and the cytosolic level of hnRNPA1 in DTT treated cells by western blotting experiments (Figure 6A). Results showed that hnRNPA1 level in the cytoplasm increased after 6 h of treatment with increasing concentrations of DTT, while the amount of hnRNPA1 in the nuclei decreased (Figure 6A), and the total hnRNPA1 level was not affected by the DTT treatment. To decipher the role of hnRNPA1 on the ER stress-mediated induction of *DLL4* IRES activity, cells were co-transfected with the bicistronic constructs and either with scrambled or hnRNPA1-specific siRNA and ER stress was induced by DTT treatment. hnRNPA1 knockdown was efficient but did not impair eIF2α phosphorylation after induction of ER stress by DTT (Figure 6B). While hnRNAP1 expression inhibition had no effect on *EMCV* IRES activity, it reduced *DLL4* IRES activity during DTT-induced ER stress, compared to scramble siRNA transfected cells (Figure 6C). The data presented are consistent with a model in which *DLL4* IRES activity is governed, at least in part, by the cellular IRES trans-acting factor hnRNPA1.

## 3. Discussion

Translational initiation is a well-established and crucial regulatory step of gene expression, allowing reprogramming of protein synthesis in response to environmental changes. Under stress conditions, the maintenance of routine translation machinery would be deleterious. Hence the synthesis of “housekeeping” proteins is paused in stressed cells whereas the translation of a pool of proteins necessary for the adaptive stress response is maintained, via alternative mechanisms of translational initiation. To bypass the stress-mediated inhibition of cap-dependent translation, more than 100 mammalian mRNAs harbor internal ribosome entry site (IRES) elements in their 5′-UTRs that mediate internal initiation of translation. These mRNA include many mRNA encoding proteins strongly involved in angiogenesis like VEGF-A, FGF-2, FGF-1A, VEGF-C, PDGF or TSP1 [48]. In this study we show that a *bona fide* IRES element is present in the 5′-UTR of *DLL4* mRNA encoding the DLL4 protein that regulates angiogenesis during development, but also in pathological conditions, such as cancer [16,20,26,49]. Interestingly, as has already been shown for the *VEGF* and *FGF-2* IRESs [2,4], the *DLL4* IRES is activated under stress conditions including hypoxia and ER stress. This suggests that DLL4 remains efficiently translated during stress despite the substantial global inhibition of cap-dependent protein translation. We have also shown the significance of the PERK kinase in regulating stress-induced IRES-dependent translation of DLL4. Taken together our data suggest that DLL4 is translated under ER stress conditions despite phosphorylation of the major PERK substrate, eIF2α. It was previously demonstrated that PERK signaling is crucial for determining the growth and angiogenesis of specific tumors [50,51]. For example, UPR induced by glucose deprivation increases VEGF expression in human tissues in a PERK-dependent manner [52]. Moreover, tumors derived from K-Ras– transformed Perk−/− MEFs were found to be smaller than those derived from MEFs with an intact integrated stress response, because targeting PERK signaling disrupts angiogenic signals and prevents the appropriate organization and maturation of functional vessels [51].

This finding has biological relevance because several components involved in the same mechanism are translated in an IRES-dependent manner, providing selective co-regulation under stress conditions (Figure 7). Indeed, during tumor progression, the stress area encompasses both the tumor and its microenvironment. The growing neovessels that are inefficiently perfused, and more particularly TIP cells which are furthest away from the circulating blood, are also subjected to stress due to the unfavorable environment (hypoxia, glucose or amino acid starvation, acidosis). These adverse conditions are known to induce ER stress leading to eIF2α phosphorylation and thus to activation of a network of genes dependent on ER stress. Many genes whose product are involved in angiogenesis, including *Vegf-a*, *Fgf-2*, *Hif1α* and *Dll4*, and expressed either by tumoral cells or by the microenvironment, contain IRES element allowing the maintenance of an efficient translation of these mRNAs under stressful conditions while cap-dependent initiation is compromised (Figure 7). A number of other reports have demonstrated that IRES-dependent translation is driven by ER stress [4,40,53,54,55]. However, our results clearly show the link between the UPR, more precisely PERK/ eIF2α signaling, and IRES-dependent translation of angiogenesis related genes. The physiological relevance of IRES-dependent translation mechanisms of non-viral mRNA is poorly documented in the literature and the precise mechanism of how stress signals downstream of eIF2α phosphorylation are transduced to IRES elements remains unknown.

It was previously largely described that upstream open reading frames (uORFs) are key regulators of mRNA translation upon eIF2α phosphorylation. In the case of stress-induced eIF2α phosphorylation, it was proposed that after having translated the uORF, recharging of the ribosome with active initiation factors (including the ternary complex eIF2.GTP.met-tRNA) is the limiting step for reinitiating translation [56]. This model account for the observed increased translation efficiency of the downstream open reading frame when the intercistronic region is longer. The hypothesis is that when the region to be scanned is long, the ribosome would have enough time to reacquire reinitiation factors before encountering the downstream initiation codon. Nevertheless, *DLL4* 5′-UTR which is highly GC rich, doesn’t contain uORF.

Moreover, secondary structures, which could be stabilized by specific protein binding, could slow down scanning of the 43S pre-initiation complex and by this way increase translation reinitiation efficiency [57]. All the cellular IRESs already characterized used the so-called “land and scan” mechanism to initiate translation. The 40S subunit associates with the IRES upstream of the initiation codon, and then scan the mRNA in a 5′-to-3′ direction until start codon recognition occurs [58,59,60]. The presence in IRES elements of extensive secondary structure, representing a significant barrier slowing down the ribosome scanning, could explain the need of IRES mediated translation under stress conditions when eIF2α is phosphorylated. We can also postulate that the presence of pause sites within the IRES sequence scanned by the ribosome, will allow it to reacquire active initiation factors and to initiate efficiently the translation at the downstream AUG. This hypothesis is supported by the fact that for *EMCV* IRES which is insensitive to ER stress, there is good evidence that the 40S ribosomal subunits associated with initiation factors bind directly to the initiating AUG without any scanning requirement [61]. Indeed, it was demonstrated that eIF2 was necessary for *EMCV* mRNA translation [62] and that either treatment stimulating PKR (RNA-dependent protein kinase R, one of the four known eIF2α kinases) or activation of PKR alone suffices to block EMCV translation [63,64] confirming that *EMCV* IRES activity was inhibited upon eIF2α phosphorylation. 

Emerging evidence has shown that the ribosome itself can play a crucial role in the specialized translation of specific subsets of mRNAs harboring specific *cis*-regulatory elements. Ribosomal biogenesis involves hundreds accessory factors and requires transcriptional and post-transcriptional steps which are timely and spatially regulated. It is thus not surprising that alteration of ribosome biogenesis is associated with dysregulation of translational efficiency. During hypoxic stress for example, cells maintained viability by restricting ribosomal biogenesis, the most energy-demanding cellular process [65]. We can hypothesize that cap-independent translation through the use of IRES elements is a mechanism, which is favored by low ribosome content during stress conditions. In the same vein, recent evidence suggests that ribosomal proteins themselves may function to recognize specific IRES elements. For example RPS19 and RPL11 regulate IRES-dependent translation of BCL-2-associated athanogene (*BAG1*) and cold shock domain containing protein E1 (*CSDE1*) [66]. 

Both the ribosomal RNA and proteins are heavily modified in mature ribosomes leading to ribosome heterogeneity that could be an important mechanism regulated during stress conditions. It was demonstrated that both 2′-O-methylation and hypo-pseudouridylation of rRNA is associated with impaired IRES-dependent translation [67,68], and that ribosomes bound to the HCV IRES have a different methylation pattern from ribosomes bound to host mRNAs, indicating a role of methylated ribosomal proteins in IRES-mediated translational control [69]. Moreover, in mouse and human cells expressing a hypomorphic form of the pseudouridine synthase dyskerin, the translation of some IRES-containing mRNAs was impaired [68,70,71]. Likewise, yeast expressing a catalytically inactive form of Cbf5, the homolog of human dyskerin, are also deficient in IRES dependent translation initiation [72]. However, it was also shown that dyskerin depletion increases VEGF mRNA internal ribosome entry site-mediated translation [73], indicating that cell type or genetic and environmental factors most certainly could influence the degree of implication of dyskerin in the translation of IRES containing mRNA. Given that hypoxic stress decreases dyskerin expression level [74], it would be of great interest to investigate the effect of ER stress or hypoxia on both ribosomal protein expression or rRNA modifications including 2′-O-methylation and pseudouridylation. Moreover, it will also be interesting to investigate whether ribosomal proteins promote specialized translation of IRES-containing mRNAs, either directly or through ITAFs. Indeed, many studies showed that ITAFs are able to stabilize the adequate RNA conformation allowing ribosome recruitment [75]. ITAFs are responsible for sensing changes in cellular metabolism and influence IRES activity. Moreover, subcellular distribution of many of them is modulated by stress [43]. Thus, both the expression level and localization of ITAFs could finely regulate the expression of IRES containing mRNAs. The results presented in this study suggest that the hnRNP A1 is an ITAF that increases the *DLL4* IRES activity in stress conditions, and that this regulation is dependent on the nucleo–cytoplasmic relocation of hnRNP A1 upon ER stress. The same mechanism was already described for the cap independent translation of the transcription factor SREBP-1 (sterol-regulatory-element binding protein 1) during ER stress [75]. Nevertheless, we did not observe a complete inhibition in IRES-mediated translation after knockdown of hnRNP A1. The main reasons are the knockdown efficiency of hnRNP A1 by the siRNA (hnRNP A1 protein is abundant in the cell) and the potential presence of other ITAFs. Further investigation would be necessary to completely understand IRES-mediated translation of *DLL4* mRNA.

## 4. Materials and Methods

### 4.1. Plasmid Constructions and Viral Production

The ER transmembrane and cytoplasmic domains of the human PERK protein were cloned from human cDNA using the PERKHALZ and PERK reverse primers (Table 1). Signal peptides and leucine zipper (LZ) peptides were merged with the PERK domains using the primers shown in Table 1. 

Sequences were confirmed by DNA sequencing. Purified PCR products were digested with KpnI and XhoI restriction enzymes and inserted into the pTRIPz-TRE-Tigh plasmid (Open Biosystems, Dharmacon Lafayette, CO, USA) to generate the pL-TRE-PLZ expression vectors. Viruses were produced by transfecting HeLa cells (5.0 × 10^4^) with the JetPEI transfection reagent (Polyplus Transfection, Illkirch, France), according to the manufacturer’s instructions. 

Constructs containing the *VEGF-A*, *FGF-1A* and *EMCV* IRESs have been previously described [4,37]. *DLL4* 5′-UTR was amplified by PCR using Phusion High-Fidelity (New England Biolabs, Evry, France) polymerase using the primers DLL4+1-F and DLL4 ATG-R. PCR product was subcloned into the pCR™-Blunt shuttle vector (Thermo Fisher Scientific, Dardilly, France), which was sequenced and then cloned into a bicistronic vector using the SpeI and NcoI restriction sites to give the pCRD4L vector. Constructs containing the *VEGF-A* IRES A, B and the full *VEGF* 5′-UTR (namely pTCRVL) upstream of a tetracycline responsive promoter have been previously described [2]. After digestion of the PCRD4L by XbaI and PacI, the cassette LucR-5′-DLL4-LucF was inserted in the vector pTCRVL also digested by XbaI and PacI, thus replacing the VEGF sequence by the DLL4 one.

### 4.2. Cell Culture and Transfection

HeLa cells, NIH3T3 and both wild-type and eIF2αS51A MEF cells (kindly provided by P. Fafournoux from INRA, Unité de nutrition Humaine, France) were cultivated in DMEM media (Sigma Aldrich, Saint-Quentin Fallavier France) supplemented with 10% FCS, 1% glutamine and antibiotics. Cells were transfected using the JetPEI transfection reagent (Polyplus Transfection), according to the manufacturer’s instructions. 24 hours after transfection with the bicistronic constructs, cells were treated with DTT, the PERK activator CCT020312 (Merck Millipore, Fontenay sous Bois, France) or the PERK inhibitor GSK2606414 (Merck Millipore) for various concentrations, and were then harvested. For hypoxia, cells were incubated at 37 °C at 1% O_2_.

Human Umbilical Vein Endothelial Cells (HUVEC) pooled from 6 donors were prepared by digestion of umbilical veins with 0.1 g/L collagenase A (Roche, Meylan, France) and cultivated in the specific Endothelial Cell Growth Medium 2 (PromoCell, Heidelberg Germany). Cells were transfected using the JetPEI-HUVEC transfection reagent (Polyplus Transfection), according to the manufacturer’s instructions.

### 4.3. Total RNA extraction and RT-PCR

Total RNA was purified using the TRI Reagent solution (Applied Biosystems, Thermo Fisher Scientific, Dardilly, France)). Reverse transcription was carried out with 1 µg of total RNA using a RevertAid First Strand cDNA Synthesis Kit (Thermo Fisher Scientific, Dardilly, France) with random hexamers, according to the manufacturer’s instructions. The resulting cDNA was amplified by PCR for 30 cycles using Phusion High-Fidelity DNA Polymerase, 2× mix (New England Biolabs) and specific primers (Table 1).

### 4.4. Western Blot Analysis

Western blotting was performed as previously described [76]. PERK, IRE1, ATF4, eIF4E-BP1, eIF2α and p-eIF2α were immunodetected using rabbit anti-human monoclonal antibodies (Cell Signaling, Danvers, MA, USA, dilution 1:1000) as the primary antibody, and peroxidase-conjugated sheep anti-rabbit (Cell Signaling Technology (Leiden, The Netherlands), dilution 1:5000) as secondary antibody. HA-tagged proteins and HIF-1α were detected using the mouse monoclonal antibodies clone HA-7 (Sigma Aldrich) and Clone 54/HIF-1α (BD Biosciences, Le Pont de Claix, France) respectively, and peroxidase-conjugated horse anti-mouse (Cell Signaling Technology, dilution 1:5000) as secondary antibody.

Protein signals were normalized using either total eIF2α or an anti-β-actin monoclonal antibody (AC-15, Sigma-Aldrich (Saint-Quentin Fallavier France), dilution 1:10,000). Signals were detected using the Clarity chemiluminescence kit (Bio Rad, Marnes-la-Coquette, France). Nuclear and cytoplasmic extracts were obtained by using NE-PER™ Nuclear and Cytoplasmic Extraction Reagents (Thermo Fisher Scientific, Dardilly, France) according to the manufacturer’s protocol.

### 4.5. RNA Interference

RNA interference-mediated gene knockdown was achieved by transfecting 5 nM of ATF6-, IRE1α-, PERK-, and ATF4-targeting or duplex control siRNA (ON-TARGETplus SMARTpool, Dharmacon, Lafayette, CO, USA) into HeLa cells using Interferin (Polyplus Transfection), according to the manufacturer’s protocol. After 48 hours, cells were co-transfected (3 µg plasmids and 5 nM siRNA) with Lipofectamine 2000 (Invitrogen, Cergy-Pontoise, France), and harvested 24 hours later for protein and RNA analyses.

### 4.6. Luciferase Activity

For reporter vectors, luciferase activities were performed using Dual-Luciferase Reporter Assay (Promega, France). The quantification of Renilla and Firefly Luciferase activities was achieved 48 h after transfection with a luminometer (TriStar² LB 942 Modular Multimode Microplate Reader, Berthold, Versailles, France), according to the manufacturer’s instructions.

### 4.7. Statistical Analyses

All experimental data are expressed as mean ± SEM and statistical significance was evaluated using Student’s *t*-test. Differences were considered significant at values of *p* < 0.05 (* *p* < 0.05; ** *p* < 0.01; *** *p* < 0.001).

## 5. Conclusions

In conclusion, DLL4 has evolved an IRES element that allows the enhancement of its translation under ER stress, a condition known to be activated during tumor development when the expression of this protein is crucial. The presence of IRES elements in many genes involved in angiogenesis and expressed in stress conditions including *Fgf-2*, *vegf-a* or *Hif1* α (Figure 7) and *Dll4* suggest that IRESs function as cis-acting regulons during ER stress. In addition, we have shown the significance of the PERK kinase in regulating stress-induced IRES-dependent translation of this mRNA. These observations suggest that PERK could be a useful druggable target to control angiogenesis, possibly even locally, following an ischemic disorder or in cancers.

## Figures and Tables

**Figure 1 cancers-11-00142-f001:**
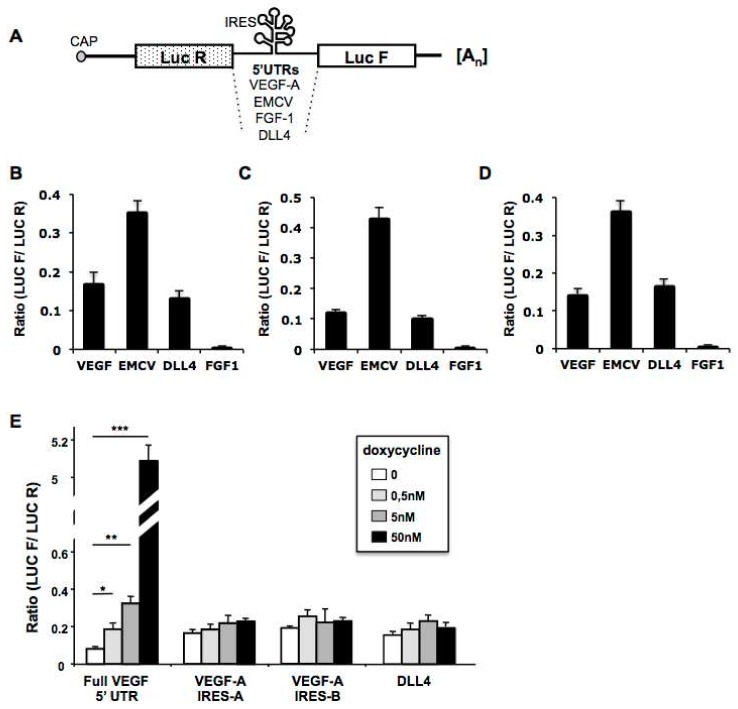
*DLL4* 5′-UTR contains an internal ribosome entry site. (**A**) Schematic representation of bicistronic constructs. IRESs cloned within the inter-cistronic region were either viral (*EMCV*) or cellular (*VEGF-A* IRESA, *VEGF-A* IRESB, *FGF-1*, *DLL4*), (**B**–**D**) Analysis of the *DLL4* IRES activity in transiently transfected (**B**) HeLa, (**C**) HUVEC or (**D**) NIH3T3 cells. 48h after transfection IRES activity was determined by calculating the LucF/LucR ratio. *DLL4* IRES activity was compared to that of the cellular IRES-A of *VEGF-A*, the viral *EMCV* IRES or the *FGF-1A* IRES, known to be highly tissue and cell line specific. (**E**) HeLa Tet off cells were transfected with TET sensitive bicistronic constructs containing the full 5′-untranslated region of *DLL4*. At 2 h prior to transfection, cells were treated with 0.5 nM, 5 nM or 50 nM doxycycline (Dox). Forty-eight hours after transfection, luciferase activities were measured as described. As positive controls, the *VEGF-A* full 5′-UTR (containing a cryptic promoter) was introduced in the intercistronic region and only the *VEGF-A* IRES A or B as negative controls (sequences without cryptic promoter). Data are means ± SEM from 3 independent experiments in duplicates, * *p* < 0.05, ** *p* < 0.01, *** *p* < 0.001.

**Figure 2 cancers-11-00142-f002:**
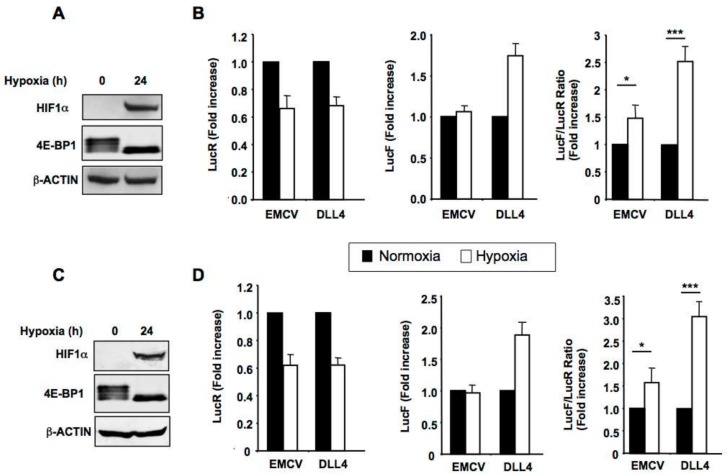
The *DLL4* IRES is activated by hypoxia. HeLa (**A**,**B**) or HUVEC endothelial cells (**C**,**D**) cells were transfected by bicistronic constructs containing either the *EMCV* or the *DLL4* IRES and submitted to 24 h hypoxia. (**A**,**C**) Hypoxia was confirmed through verifying both expression induction of HIF-1α and 4E-BP1 dephosphorylation (lower band visible at 24 h of hypoxia) by western blotting. β-ACTIN was used as a loading control. (**B**,**D**) Relative luciferase activities LucR, LucF or LucF/LucFR ratio (fold increase) under normoxia (black bars) and hypoxia (white bars) in HeLa (**B**) or HUVEC (**D**) cells. Results represent the means of three independent experiments (±SEM), * *p* < 0.05, *** *p* < 0.001.

**Figure 3 cancers-11-00142-f003:**
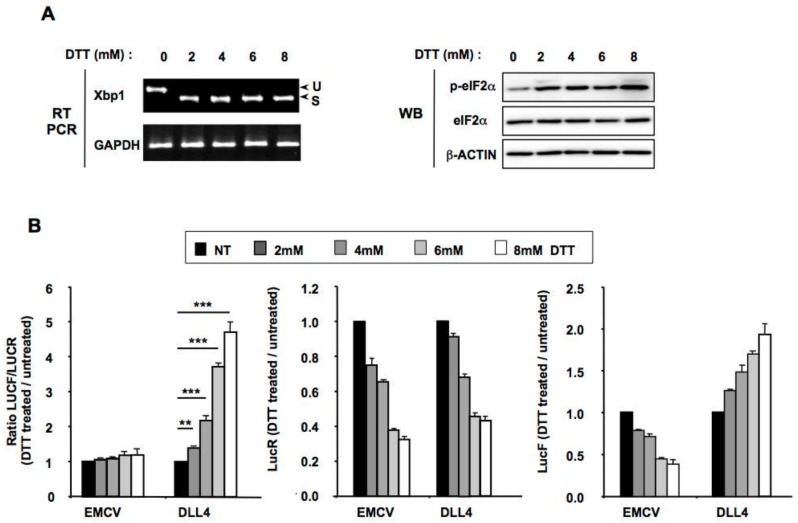
The *DLL4* IRES is activated by ER stress. HeLa cells were transfected by bicistronic constructs containing either the *EMCV* or the *DLL4* IRES and treated or not for 6 h with increasing concentrations of DTT. (**A**) ER stress induction was verified by monitoring both cytoplasmic *XBP1* splicing by RT-PCR and eIF2 phosphorylation by western blotting (**B**) Relative luciferase activities (LucR, LucF or LucF/LucR ratio) after ER stress induction by treatment with increasing amounts of DTT for 6 h. IRES activities were determined by calculating the LucF/LucR ratios and are expressed as fold change versus untreated cells. The results represent the means of three independent experiments (±SEM), ** *p* < 0.01, *** *p* < 0.001.

**Figure 4 cancers-11-00142-f004:**
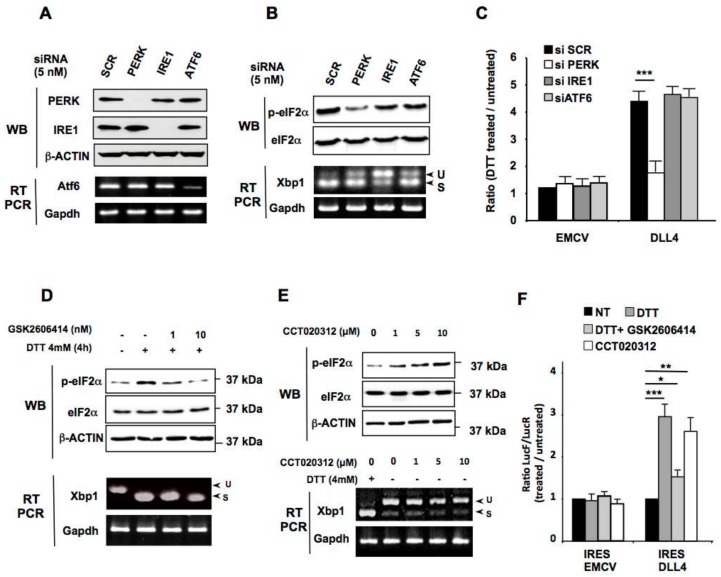
Role of three UPR sensors in *DLL4* IRES translation activation by ER Stress. (**A**) HeLa cells were transfected with PERK, ATF6, IRE1 or control (Scr) siRNA. The knockdown efficiency of the targeted transcripts was determined by western blotting with PERK, IRE1 and β-ACTIN antibodies, and by semi-quantitative RT-PCR against ATF6 and GAPDH as a control. (**B**) Western blot analysis of phosphorylated eIF2α and total eIF2α and RT-PCR analysis of *XBP1* splicing performed after transfection of HeLa cells either with siRNA specific for PERK, ATF6, or IRE1 or with control siRNA (Scr), after treatment with DTT. (**C**) Relative IRES activities in HeLa cells treated with DTT/control after co-transfection with siRNA specific for PERK, ATF6 or IRE1, or with control siRNA (scr) and DLL4 or EMCV bicistronic vectors. (**D**) Western blot analysis of phosphorylated eIF2α and total eIF2α and RT-PCR analysis of *XBP1* splicing performed after transfection of HeLa cells with DLL4 or EMCV bicistronic vectors, after treatment with DTT and increasing concentration of the PERK inhibitor GSK2606414. (**E**) Western blot analysis of phosphorylated eIF2α and total eIF2α and RT-PCR analysis of *XBP1* splicing performed after transfection of HeLa cells with DLL4 or EMCV bicistronic vectors, after treatment with increasing concentration of the PERK activator CCT020312. (**F**) Relative IRES activities in HeLa cells treated with DTT/control after transfection with DLL4 or EMCV bicistronic vectors and treatment with PERK activator (CCT020312) or inhibitor (GSK2606414). The results represent the means of three independent experiments (±SEM), * *p* < 0.05, ** *p* < 0.01, *** *p* < 0.001.

**Figure 5 cancers-11-00142-f005:**
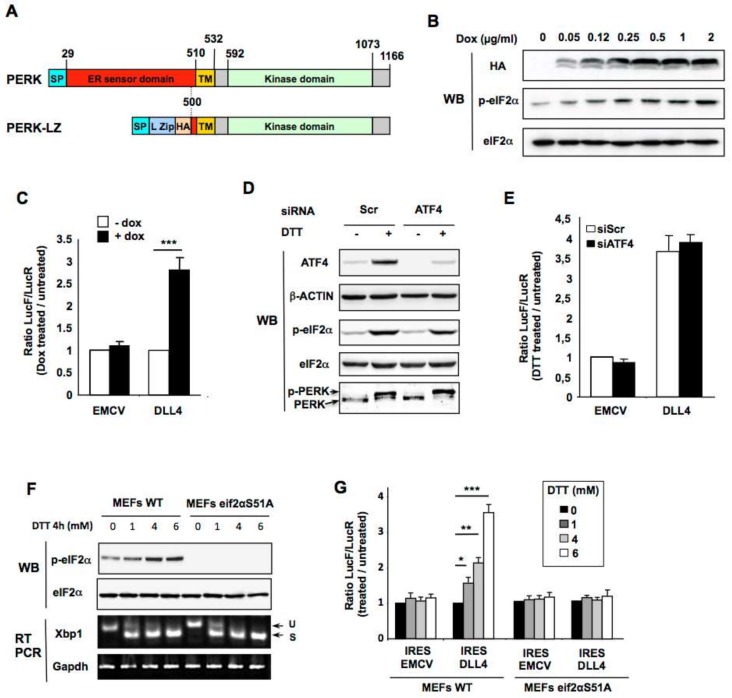
PERK kinase activity stimulates *DLL4* IRES-mediated translation in a phosphorylated eIF2α–dependent manner. (**A**) Linear schematic representation of PERK and PERK-LZ showing the locations of the major functional domains (SP = Signal Peptide, L zip = Leucine Zipper, HA = HA Tag). Numbers indicate amino acid positions. (**B**) Western blot analysis of doxycycline-induced-PERK-LZ (HA) expression and eIF2α phosphorylation after 48 h treatment with increasing amounts of doxycycline. (**C**) Relative IRES activities in PERK-LZ expressing HeLa cells treated with 1 µg/mL doxycycline /control after transfection of EMCV or DLL4 bicistronic vectors. Means ± SEM are shown, *** *p* < 0.001. (**D**) Western blot analysis of ATF4, total and phosphorylated eIF2α, as well as total PERK in HeLa cells, after transfection with ATF4 or scramble (Scr) siRNA and treatment with DTT. β-ACTIN was used as a loading control. (**E**) Relative IRES activities in HeLa cells treated with DTT/control, after cotransfection with the EMCV or DLL4 bicistronic vectors and with either siRNA specific for ATF4 or control siRNA (Scr). Results represent the means of three independent experiments ± SEM. (**F**) Wild-type (WT) and eIF2αS51A MEFs transfected with the bicistronic LucR-IRES-LucF vectors and treated with increasing concentrations of DTT. ER stress induction was verified by monitoring eIF2α phosphorylation by western blot and cytoplasmic *XBP1* splicing by RT-PCR. (**G**) Relative IRES activities were determined as previously described in WT (left) and eIF2αS51A (right) MEFs. Results represent the means of three independent experiments ± SEM, * *p* < 0.05, ** *p* < 0.01, *** *p* < 0.001.

**Figure 6 cancers-11-00142-f006:**
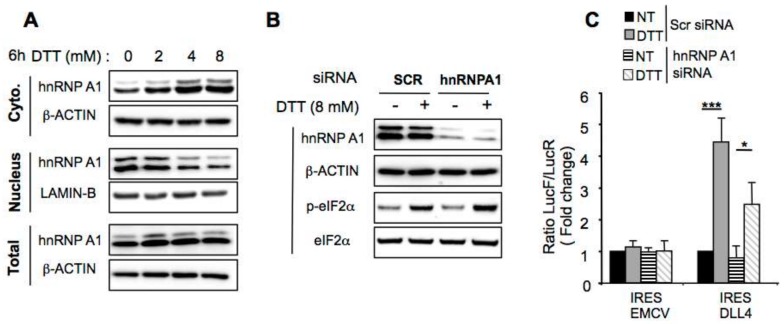
hnRNPA1 modulates *DLL4* IRES activity during ER stress. (**A**) Western blot analysis of hnRNPA1 in HeLa nuclear, cytoplasmic or total extract after treatment with increasing concentrations of DTT. β-ACTIN was used as a loading control for cytoplasmic and total extracts and LAMIN-B for nuclear extract. (**B**) Western blot analysis of hnRNPA1, total and phosphorylated eIF2 in HeLa cells, after transfection with hnRNPA1 or scramble (Scr) siRNA and treatment with DTT. β-ACTIN was used as a loading control. (**C**) Relative IRES activities in HeLa cells treated with DTT/control, after cotransfection with the EMCV or DLL4 bicistronic vectors and with either siRNA specific for hnRNPA1 or control siRNA (Scr). Results represent the means of three independent experiments ± SEM. * *p* < 0.05, *** *p* < 0.001.

**Figure 7 cancers-11-00142-f007:**
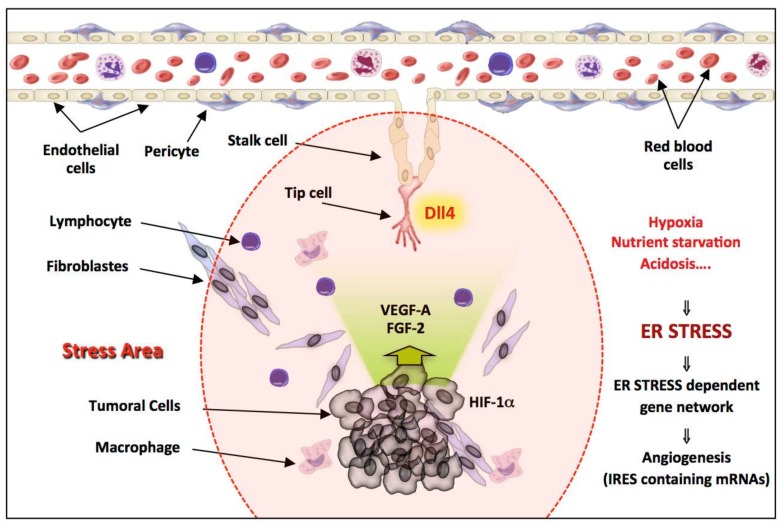
Schematic model of the network of gene expression co-regulation by stress during tumoral progression. During tumor progression, the stress zone encompasses the growing tumor, but also its microenvironment. The neo vessels and more particularly the Tip cells, present at the extremity and which guide the neo vessels towards the tumor, are located in this unfavorable microenvironment. Hypoxia, nutrient starvation and acidosis will irremediably induce the accumulation of unfolded protein in the reticulum of cells located in this area, leading to ER stress and UPR activation. Thus, in addition to transcriptional regulations, the activation of the PERK pathway will induce the co-regulation of an UPR dependent gene network containing IRES elements, revealing a translational regulon in which the synthesis of a cohort of angiogenic related genes is activated in response to ER stress. The fine-tuning of gene expression allows an efficient control of angiogenesis, which is a highly regulated process.

**Table 1 cancers-11-00142-t001:** Oligonucleotide sequences.

Primer Name	Sequence 5′→3′
PERKHALZ	CAGCTCAAGACGCGTTTCGAATACCCATACGATGTTCCTGACTATGCGAGATTCCTCGACAACCCACA
PERK rev	TTCTCGAGTATCGATTTACTAATTGCTTGGCAAAGGGC
SP	AAACTAGTGCCATGGCTCCGGCCCGGCGGCTGCTGCTGCTGCTGACGCTGCTGCTGCCCGGCCT
SP-LZ 1	CACTTTCTCTTCCAGGCGCGATGTGCTGGTACTTCCAAAAATCCCGAGGCCGGGCAGCAGCAGCGT
SP-LZ 2	CGCGCCTGGAAGAGAAAGTGAAGACCCTCAAGAGTCAGAACACGGAGCTGGCGTCCACGGCGAGC
LZ	TTCGAAACGCGTCTTGAGCTGCGCCACCTGCTCGCGCAGCAGGCTCGCCGTGGACGCCAGCTC
XBP1-F	CTGGAACAGCAAGTGGTAGA
XBP1-R	CTCCTCCAGGCTGGCAGG
DLL4+1-F	AAACTAGTGCTGCGCGCAGGCCGGGAACACG
DLL4 ATG-R	AAAACCATGGCCCCTCGGGCGTCGCTCTCTC
GAPDH-F	CAAGGTCATCCATGACAACTTTG
GAPDH-R	GTCCACCACCCTGTTGCTGTAG
ATF6-F	GGGAGACACATTTTATGTTGTGTC
ATF6-R	GGTTTGATTCCTCTGCTGATCTCG

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
