# Peer review of "The PERK Branch of the Unfolded Protein Response Promotes DLL4 Expression by Activating an Alternative Translation Mechanism"

_cancers, 2019, doi:10.3390/cancers11020142_

Round 1
Reviewer 1 Report
This paper by Jaud et al describes the role of PERK in promoting expression of DLL4 through IRES. The manuscript is a very thorough investigation of this alternative mechanism of translation of DLL4, and has used pharmacological approaches and elegant genetic approaches to demonstrate this regulatory mechanism. The manuscript is generally very well written. I have some relatively minor comments aimed to improve the manuscript.
1. Figure 6A – the authors should indicate if one of the bands on the hnRNP A1 blots are non-specific, and indicate which one with an asterix. Additional controls for subcellular fractions should be shown, ie lack of actin in the nuclear fraction and lack of laminB in the cytosolic fraction.
2. The authors should include statistical significance of their results, particularly for Figure 6C – statistical analysis of these data should be included
3. The authors should discuss the role of hnRNP A1 in the discussion.
4. Line 530 – It should be possible to perform at least Student t test analysis with n=3. I don’t understand the author’s statement ‘There is no power to perform statistical comparisons between groups.’
Minor typographical errors etc
5. Line 57: dependent on, not dependent of
6. Introduction: several of the paragraphs are very short and could be combined for better flow.
7. Figure 1D should be referred to in the text around line 144
8. Formatting of the references should be checked, eg in Reference 4 mrnas should be mRNAs
9. Figure 5F, please move the labelling slightly so that the GAPDH label is fully visible
10. Line 258 - 'To further sustain this…', might be better expressed as 'To complement the …'
11. Line 282 – bicistronic is misspelt
12. Line 290 – use the abbreviation eIF2alpha
13. Figure 7 should not have an accent on the first e.
14. Line 381 – remove …. after acidosis
15. Line 389 – ‘more precisely the PERK/ eIF2α couple’ could be better expressed as ‘more precisely PERK/eIF2a signalling’
16. Line 398 – there should be a space before [53]
17. The paragraph starting at line 420 could be combined with the two paragraphs that follow, for better flow. Similarly the last two paragraphs in the discussion could be combined.
Author Response
Please find enclosed a revised version of our manuscript entitled “The PERK branch of the unfolded protein response promotes DLL4 expression by activating an alternative translation mechanism” by M.Jaud, C. Philippe, et al., corresponding to manuscript “cancers-410166”
We thank the reviewers for their thorough and critical reading of the manuscript, and though their overall opinion was positive their specific remarks were very useful for improving the manuscript. We have responded to each of their criticisms below. To address the reviewers’ comments we have added many changes in the text.
We hope this new version meets with everyone’s approval and thank you for your consideration.
Yours sincerely,
Christian Touriol
Reviewer 1
Comments and Suggestions for Authors
This paper by Jaud et al describes the role of PERK in promoting expression of DLL4 through IRES. The manuscript is a very thorough investigation of this alternative mechanism of translation of DLL4, and has used pharmacological approaches and elegant genetic approaches to demonstrate this regulatory mechanism. The manuscript is generally very well written. I have some relatively minor comments aimed to improve the manuscript.
1. Figure 6A – the authors should indicate if one of the bands on the hnRNP A1 blots are non-specific, and indicate which one with an asterix. Additional controls for subcellular fractions should be shown, ie lack of actin in the nuclear fraction and lack of laminB in the cytosolic fraction.
Both bands on the blot are specific. Indeed, two major alternatively spliced isoforms were already described (NM_002136.3 → NP_002127.1 heterogeneous nuclear ribonucleoprotein A1 isoform a, and NM_031157.3 → NP_112420.1 heterogeneous nuclear ribonucleoprotein A1 isoform b, containing an alternate in-frame exon compared to variant 1, resulting in a longer protein). Thus the hnRNP A1 (D21H11) Rabbit mAb Antibody from cell signaling used in this study reveal a doublet at about 35/40 kDa in HeLa cells.
As indicated in the material and method, nuclear and cytoplasmic fractions were obtained by using the commercial KIT NE-PER™ Nuclear and Cytoplasmic Extraction Reagents (Thermo Fisher). Nevertheless, to control the efficiency of both nuclear and cytoplasmic protein extraction, and to demonstrate that the DTT treatment has no effect on the fractionation efficiency, a western blot was now added in a Supplementary figure 2 and we modified the text accordingly.
2. The authors should include statistical significance of their results, particularly for Figure 6C – statistical analysis of these data should be included
In each case, the data presented are the results of at least 3 experiments. All experimental data are expressed as mean ± SEM.
We decided in the first version of the paper to not include statistical significance in our results, (and thus to precise “For all these in vitro experiments, only descriptive statistics were performed because of small sample size (at least n=3). There is no power to perform statistical comparisons between groups”) because biostatistician has warned us that the problem with low sample size (n=3) was with regard to the power of the test. Indeed, some people argue that t-test procedure is not advisable with n < 10, but others consider that a sample size of n=3 is sufficient for the t test .
However, upon the reviewer’s request, we now included statistical significance in our results.
3. The authors should discuss the role of hnRNP A1 in the discussion.
A new paragraph at the end of the discussion section was added to discuss the role of hnRNPA1 in the cap independent translational regulation.
4. Line 530 – It should be possible to perform at least Student t test analysis with n=3. I don’t understand the author’s statement ‘There is no power to perform statistical comparisons between groups.’
See answer to comment number 2
Minor typographical errors etc
All the minor typographical errors were corrected in the new version
5. Line 57: dependent on, not dependent of
6. Introduction: several of the paragraphs are very short and could be combined for better flow.
7. Figure 1D should be referred to in the text around line 144
8. Formatting of the references should be checked, eg in Reference 4 mrnas should be mRNAs
9. Figure 5F, please move the labelling slightly so that the GAPDH label is fully visible
10. Line 258 - 'To further sustain this…', might be better expressed as 'To complement the …'
11. Line 282 – bicistronic is misspelt
12. Line 290 – use the abbreviation eIF2alpha
13. Figure 7 should not have an accent on the first e.
14. Line 381 – remove …. after acidosis
15. Line 389 – ‘more precisely the PERK/ eIF2α couple’ could be better expressed as ‘more precisely PERK/eIF2a signalling’
16. Line 398 – there should be a space before [53]
17. The paragraph starting at line 420 could be combined with the two paragraphs that follow, for better flow. Similarly the last two paragraphs in the discussion could be combined.

Reviewer 2 Report
This is a very interesting work, performed in continuation of the previous works by the authors. I detected only small written english mistakes (e.g. lines 18 and 19, 37), formatting problems (e.g. legends of figures 4 and 7 and line 492), absence of abbreviations (in the abstract). For me, the main problem is the number of repetitions performed to ensure the statistical confidence of this work and to support the conclusions. In this context, the number of repetitions is not clear. The authors referred "three independent experiments" several times, however, in section 4.7 it seems that only a total n=3 was effected and the statistical confidence on the results is not established (Students-t test?).
Author Response
Please find enclosed a revised version of our manuscript entitled “The PERK branch of the unfolded protein response promotes DLL4 expression by activating an alternative translation mechanism” by M.Jaud, C. Philippe, et al., corresponding to manuscript “cancers-410166”
We thank the reviewers for their thorough and critical reading of the manuscript, and though their overall opinion was positive their specific remarks were very useful for improving the manuscript. We have responded to each of their criticisms below. To address the reviewers’ comments we have added many changes in the text.
We hope this new version meets with everyone’s approval and thank you for your consideration.
Yours sincerely,
Christian Touriol
Reviewer 2
Comments and Suggestions for Authors
This is a very interesting work, performed in continuation of the previous works by the authors. I detected only small written english mistakes (e.g. lines 18 and 19, 37), formatting problems (e.g. legends of figures 4 and 7 and line 492), absence of abbreviations (in the abstract). For me, the main problem is the number of repetitions performed to ensure the statistical confidence of this work and to support the conclusions. In this context, the number of repetitions is not clear. The authors referred "three independent experiments" several times, however, in section 4.7 it seems that only a total n=3 was effected and the statistical confidence on the results is not established (Students-t test?).
In each case, the data presented are the results of at least 3 experiments. All experimental data are expressed as mean ± SEM.
We decided in the first version of the paper to not include statistical significance in our results, (and thus to precise “For all these in vitro experiments, only descriptive statistics were performed because of small sample size (at least n=3). There is no power to perform statistical comparisons between groups”) because biostatistician has warned us that the problem with low sample size (n=3) was with regard to the power of the test. Indeed, some people argue that t-test procedure is not advisable with n < 10, but others consider that a sample size of n=3 is sufficient for the t test .
However, upon the reviewer’s request, we now included statistical significance in our results.
Reviewer 3 Report
In this manuscript entitled “PERK branch of the unfolded protein response promotes DLL4 expression by activating alternative translation mechanism” Jaud et al aim to demonstrate that mRNA encoding DLL4 contains an internal ribosome entry site through which DLL4 is preferentially translated under stress conditions in a PERK-dependent manner. The corollary of this is that harsh tumour microenvironments induce PERK activation which maintains DLL4 translation and thus modulates angiogenesis. PERK has been described elsewhere to regulate angiogenesis genes at the translational level, but the regulation of DLL4 in this way is a novel discovery. The authors followed a logical path and used relevant controls and methods to answer the central question of the study. However, the authors should address the following issues to improve the quality of their manuscript.
Minor Issues:
1. A diagram depicting a linear schematic representation of the bicistronic renilla/firefly luciferase construct would aid the reader in understanding how this experiment works.
2. A sentence should be added to the end of the first paragraph of section 2.1 to make it completely clear what the 70% conservation of DLL4 5’UTR suggests.
3. A sentence should be added near the start of section 2.2 to reiterate the rational for testing the regulation of DLL4 in hypoxia.
4. There is a literature describing roles for PERK in promoting angiogenesis. Some of these papers should be discussed either in the introduction or the discussion. PMID17030613 describes Perk-dependent translational regulation of angiogenesis and should be mentioned at the very least.
5. Formatting errors/suggestions:
a. Gene and mRNA names should be written in italics (example in line 179) and be capitalised appropriately (example: HIF1A not capitalised in figure 2A.) This needs to be addressed throughout the manuscript, not only for these examples.
b. Unspliced XBP1 mRNA is usually labelled as “u” or “XBP1u” but not “us”. This should be changed.
c. The labelling of proteins/genes is not consistent throughout. PeIF2α is sometimes written as P-eIF2α and as p-eIF2α. p-eIF2α is the most conventional, and should be used. si(RNA) sometimes written as Si. Inconsistencies like these need to be addressed throughout the manuscript, not only for these examples.
d. Commas are used instead of decimal points throughout the figures.
e. The text on some figures is blurry and difficult to read. The y-axis on figure 4F is an example of this.
f. The font size in inconsistent in the figure 7 legend.
g. The phospho-PERK band in figure 5D should be appropriately labelled to aid understanding of readers.
h. The acronym ER is not used consistantly. See figure 7.
i. Suggested change to the title: The PERK branch of the unfolded protein response promotes DLL4 expression by activating an alternative translation mechanism
6. Spelling errors:
a. response(s) – Line 94
b. contain(s), (i)nternal – Line 151
c. bici(s)tronic – Line 282
d. pericyte is spelled péricyte in figure 7
7. Other suggestions for improving presentation
a. Black borders surrounding all images of blots/gels would improve presentation.
b. The image quality is not consistent between figures.
c. Remove “For all these in vitro experiments” from line 529.
Author Response
Please find enclosed a revised version of our manuscript entitled “The PERK branch of the unfolded protein response promotes DLL4 expression by activating an alternative translation mechanism” by M.Jaud, C. Philippe, et al., corresponding to manuscript “cancers-410166”
We thank the reviewers for their thorough and critical reading of the manuscript, and though their overall opinion was positive their specific remarks were very useful for improving the manuscript. We have responded to each of their criticisms below. To address the reviewers’ comments we have added many changes in the text.
We hope this new version meets with everyone’s approval and thank you for your consideration.
Yours sincerely,
Christian Touriol
Reviewer 3
Comments and Suggestions for Authors
In this manuscript entitled “PERK branch of the unfolded protein response promotes DLL4 expression by activating alternative translation mechanism” Jaud et al aim to demonstrate that mRNA encoding DLL4 contains an internal ribosome entry site through which DLL4 is preferentially translated under stress conditions in a PERK-dependent manner. The corollary of this is that harsh tumour microenvironments induce PERK activation which maintains DLL4 translation and thus modulates angiogenesis. PERK has been described elsewhere to regulate angiogenesis genes at the translational level, but the regulation of DLL4 in this way is a novel discovery. The authors followed a logical path and used relevant controls and methods to answer the central question of the study. However, the authors should address the following issues to improve the quality of their manuscript.
Minor Issues:
1. A diagram depicting a linear schematic representation of the bicistronic renilla/firefly luciferase construct would aid the reader in understanding how this experiment works.
The diagram of the bicistronic renilla/firefly luciferase mRNA was added in the new figure 1A.
2. A sentence should be added to the end of the first paragraph of section 2.1 to make it completely clear what the 70% conservation of DLL4 5’UTR suggests.
As indicated at the beginning of section 2.1, DLL4 5’UTR shows fairly high conservation, with more than 70% mRNA sequence identities between 14 mammal species. Thus the sequence is 70% identical between these14 species. To clarify we added in the text the sentence “This indicates that this non-coding region may contain functional RNA structures or regulatory sequences important for the translation of the DLL4 mRNA”
3. A sentence should be added near the start of section 2.2 to reiterate the rational for testing the regulation of DLL4 in hypoxia.
A new sentence was added at the beginning of the section 2.2 to reiterate the rational for testing the regulation of DLL4 in hypoxia.
4. There is a literature describing roles for PERK in promoting angiogenesis. Some of these papers should be discussed either in the introduction or the discussion. PMID17030613 describes Perk-dependent translational regulation of angiogenesis and should be mentioned at the very least.
We apologize for mistakenly having excluded this information. A paragraph was now added in the discussion section to mention the work of Blais et al (2006), Bi et al. (2005) and Wang et al. (2012) demonstrating the role of UPR and PERK dependent translational regulation in the angiogenic process. Consistently, these 3 new references were added in the reference section.
5. Formatting errors/suggestions:
All the Formatting errors and suggestions of the reviewer have been corrected in the new version
a. Gene and mRNA names should be written in italics (example in line 179) and be capitalised appropriately (example: HIF1A not capitalised in figure 2A.) This needs to be addressed throughout the manuscript, not only for these examples.
b. Unspliced XBP1 mRNA is usually labelled as “u” or “XBP1u” but not “us”. This should be changed.
c. The labelling of proteins/genes is not consistent throughout. PeIF2α is sometimes written as P-eIF2α and as p-eIF2α. p-eIF2α is the most conventional, and should be used. si(RNA) sometimes written as Si. Inconsistencies like these need to be addressed throughout the manuscript, not only for these examples.
d. Commas are used instead of decimal points throughout the figures.
e. The text on some figures is blurry and difficult to read. The y-axis on figure 4F is an example of this.
f. The font size in inconsistent in the figure 7 legend.
g. The phospho-PERK band in figure 5D should be appropriately labelled to aid understanding of readers.
h. The acronym ER is not used consistantly. See figure 7.
i. Suggested change to the title: The PERK branch of the unfolded protein response promotes DLL4 expression by activating an alternative translation mechanism
6. Spelling errors:
All the spelling errors have ben corrected in the new version. We apologize for these mistakes.
a. response(s) – Line 94
b. contain(s), (i)nternal – Line 151
c. bici(s)tronic – Line 282
d. pericyte is spelled péricyte in figure 7
7. Other suggestions for improving presentation
We taken into consideration all the suggestion of the reviewer
a. Black borders surrounding all images of blots/gels would improve presentation.
b. The image quality is not consistent between figures.
c. Remove “For all these in vitro experiments” from line 529.
Round 2
Reviewer 2 Report
The document was improved and, in my oppinion, can be accepted after a final english improvement.